# Occurrence and Characteristics of *Serpula himantioides* Fruiting Bodies on Living Trees in Japan

**DOI:** 10.3390/jof10080572

**Published:** 2024-08-14

**Authors:** Ryusei Haraguchi, Toshihide Hirao, Toshihiro Yamada

**Affiliations:** 1The University of Tokyo Chichibu Forest, Graduate School of Agricultural and Life Sciences, The University of Tokyo, 1-1-49 Hinoda-machi, Chichibu, Saitama 368-0034, Japan; 2The University of Tokyo Forests, Graduate School of Agricultural and Life Sciences, The University of Tokyo, 1-1-1, Yayoi, Bunkyo-ku, Tokyo 113-8657, Japan

**Keywords:** fruiting body, heartwood decay, spatial distribution pattern, wood decay fungus

## Abstract

*Serpula himantioides* is a globally distributed wood decay fungus that causes heartwood decay in several tree species. We investigated the occurrence of *S. himantioides* fruiting bodies in Japan for two years and six months to characterize their biology. The fruiting bodies matured in autumn and occurred on living *Chamaecyparis pisifera*, *Chamaecyparis obtusa*, *Larix kaempferi*, and *Cryptomeria japonica* trees, as well as on dead trees and soil. Assessing three circular plots, the incidence of living trees with *S. himantioides* fruiting bodies was lowest in the plot with the most advanced heartwood decay. Furthermore, fruiting bodies occurred more frequently in the lower slope direction of the trunk. Analysis using the pair correlation function suggested that the spatial distribution pattern of living trees with fruiting bodies may change from intensive to random with heartwood decay progress. Finally, according to generalized linear and generalized linear mixed models, which were used to investigate the factors affecting the development of fruiting bodies in *C. pisifera*, *C. obtusa*, and *L. kaempferi*, no clear relationship was found between the presence or absence of fruiting bodies and heartwood decay. Thus, we suggest that fruiting bodies can occur in healthy living trees as well as in living trees in the early stages of heartwood decay.

## 1. Introduction

Root and butt rot diseases are some of the driving forces behind the spatiotemporal diversification of forests; they are common diseases that degrade timber quality, resulting in timber discoloration, structural weakness, and eventually significant economic losses [1]. Understanding their ecology and mode of infection is crucial for controlling the damage caused by wood decay fungi. For example, the important decay fungi *Armillaria* spp. are able to spread over long distances via rhizomorphs despite the absence of root contact between adjacent trees; thus, careful removal of stumps or dying root systems is advised to reduce fungal inoculum before reforestation [1]. For *Heterobasidion* spp., another important decay fungus, primary infection occurs via airborne basidiospores on fresh stumps or wounds, while secondary infection occurs by vegetative spreading from stump to tree or from tree to tree through root contacts. Consequently, the chemical treatment of stumps [2] and adjusting the timing of thinning and clear-cutting [3,4] are regarded as control measures.

*Serpula himantioides* (Fr.) P.Karst. is a wood-decay fungus that causes heartwood decay in living trees and has been found on all continents except for Antarctica [5]. Recent studies have shown that strains isolated in Japan belong to a new cryptic lineage [6]. *Serpula himantioides* can infect a wide range of woody plant species, with butt rot reported in *Larix kaempferi* (Lamb.) Carrière and *Pseudotsuga menziesii* (Mirb.) Franco in Germany [7], *P. menziesii* in Denmark [8], *Picea glauca* (Moench) Voss, *Picea mariana* (Mill.) Britton, Sterns & Poggenb., and *Abies balsamea* (L.) Mill. in Canada [9], and *A. balsamea* and *Picea rubens* Sarg. in the USA [10]. In Japan, heartwood decay caused by *S. himantioides* has been reported in *Chamaecyparis pisifera* (Sieb. & Zucc.) Endl. [11,12]. Wood decay fungi belonging to Basidiomycota are fundamentally divided into brown rot fungi, which preferentially attack and rapidly depolymerize cellulose and hemicelluloses, and white rot fungi, which can progressively degrade both carbohydrates and lignin [13,14]. Previous studies have reported that *S. himantioides* causes brown rot in the heartwood of conifer trees [7,8,9,10,11,12], and fallen trees have been reported to occur due to advanced decay (Figure 1) [7,8,11]. In addition, there were no cases of *S. himantioides* causing damage to living trees other than heart rot, such as damage to the sapwood and bark. Nevertheless, little is known about the infection and population ecology of this fungus [8].

Studying fruiting body development in wood-decay fungi is important to elucidate their infection and population ecologies. Studies conducted in Germany and Finland have demonstrated that the risk of primary infection by *Heterobasidion* increases when large-diameter decayed spruce logs are left in the forest owing to the formation of large numbers of fruiting bodies [15,16]. *Serpula himantioides* produces thin, resupinate, and brownish annual basidiocarps [5]. Previous field development of *S. himantioides* fruiting bodies has been reported on spruce logs and stumps in late autumn in Germany [7]; on stumps and windfall trees in contact with small roots and on mineral soil on their undersides in *P. menziesii* forests in September and October in Denmark [8]; and on *Abies pindrow* (Royle ex D.Don) Royle stumps and *Pinus wallichiana* A. B. Jacks (status unknown) in October in the Himalayan woodlands [17]. In Japan, a development in *C. pisifera* forests (timing unknown) [12] has also been reported.

Most of the existing reports focus on stumps or fallen trees, with little information on the occurrence of fruiting bodies on living trees. Moreover, studies on the occurrence of *S. himantioides* fruiting bodies have not been conducted spanning multiple years. Therefore, this study investigated the occurrence of *S. himantioides* fruiting bodies on living trees over a period of two years and six months. We examined the hosts and development sites, the relationship between fruiting body development and decay progression, spatial distribution patterns, and factors that influence fruiting body development. Our observations suggested that fruiting bodies may develop in living trees in the early stages of heartwood decay, as no relationship was found between fruiting body development and heartwood decay.

## 2. Materials and Methods

### 2.1. Field Survey

To identify *S. himantioides* fruiting bodies, an area of 16 km east–west and 6 km north–south within the University of Tokyo Chichibu Forest was traversed (Figure 2) from July 2021 to December 2023. When fruiting bodies were found on a tree, the forest type, tree species, and life status (living or dead) of the tree were recorded. Additionally, a portion of the fruiting body was collected to identify the species from its DNA.

We established circular plots (A, B, and C) with 30 m diameters in three coniferous plantations, where many *S. himantioides* fruiting bodies were found during the traverses (Appendix A). Plot A consisted of *C. pisifera*, while plots B and C comprised *Chamaecyparis obtusa* (Sieb. et Zucc.) Endl. and *L. kaempferi* trees (Appendix A). No previous thinning had been conducted in the plots, and no old stumps were present prior to reforestation. Color changes in the *S. himantioides* fruiting bodies within the plots were recorded once or twice a month, while basidiospores were collected by pressing cellophane tape against the mature fruiting bodies to isolate the fungi and observe their shape. A single fruiting body was randomly selected from each plot, and the color and shape of the basidiospores were recorded. Furthermore, the length and width of 20 basidiospores per fruiting body were measured. The number of trees in each plot and those with fruiting bodies were recorded for each tree species at three time points (October 2021, 2022, and 2023). The diameter at breast height (DBH) of the trees in the plots and the height of the fruiting bodies were also measured. The TruPulse360R laser rangefinder (Laser Technology Inc., Centennial, CO, USA) was used to measure the orientation of *S. himantioides* fruiting bodies, as well as the position of the standing trees in the plot. The orientation of the fruiting bodies was recorded from the center of the tree to the center of the width of the fruiting body.

The internal decay status of the living trees in the plot was investigated using the lateral impact vibration method [18,19,20,21,22,23]. Additionally, the product of the diameter (*D*) of a tree and the resonance frequency (*Fr*) by lateral impact was used as the diagnostic index (*DFr* value). For healthy trees, the following equation is true [18,19]:*DFr* = (*E/*(*ρ* · *k*))^1/2^(1)
where *D* is the trunk diameter (cm), *Fr* is the resonance frequency (Hz), *E* is Young’s modulus in the cross-sectional direction (Pa), *ρ* is the density (g/cm^3^), and *k* is the shape factor in the trunk cross-section. Because Young’s modulus generally increases with density, *E*/*ρ* is generally constant for all species in healthy trees. However, *DFr* values of damaged trees with heartwood decay have been experimentally suggested to be lower than those of healthy trees [18,19]. Therefore, heartwood decay can be identified when the measured *DFr* values are lower than the range of *DFr* values for healthy trees. The Ponta Pro Version (WORLD survey and design Co., Ltd., Shimane, Japan) was used for diagnosis of standing trees with a trunk circumference ≥ 60 cm. We used the *DFr* specified by the manufacturer for healthy trees of *C. obtusa* and *L. kaempferi*, as well as that set by Haraguchi et al. [11] for *C. pisifera*, to determine the presence of internal decay. Considering that a previous study [11] reported heartwood decay reaching up to 3 m, we conducted our diagnosis at 1.2 m above ground level.

### 2.2. Isolation and Identification of S. himantioides

To confirm that the developed fruiting bodies belonged to *S. himantioides*, genomic DNA was extracted from the collected samples using the NucleoSpin Plant II kit (Macherey-Nagel GmbH & Co. KG, Düren, Germany) following the manufacturer’s instructions. DNA yields were determined using a Qubit dsDNA HS Assay Kit (Thermo Fisher Scientific, Waltham, MA, USA), and a polymerase chain reaction (PCR) was performed using the genomic DNA template with a *S. himantioides* species-specific primer set and a thermal cycler (GeneAtlas G; ASTEC, Fukuoka, Japan). The forward and reverse primers were SHF (5′-CTCGCATCGATGAAGAAC-3′) and SHR (5′-CAAAACATTGTCTTACGACG-3′) [24], respectively. PCR was performed in a 25 μL reaction mixture containing 5 ng of template DNA, 0.25 μM each of the forward and reverse primers, 12.5 μL of 2 × Gflex PCR Buffer, and 0.5 μL of the Tks Gflex DNA polymerase (Takara Bio Inc., Shiga, Japan). The conditions for PCR with the *S. himantioides* species-specific primer set were 94 °C for 1 min, followed by 30 cycles of 10 s at 98 °C, 15 s at 60 °C, and 30 s at 68 °C, with a final extension step at 68 °C for 7 min. After PCR, agarose gel electrophoresis was performed to verify the presence or absence of bands using the Benchtop 2UV transilluminator (Analytik Jena, Upland, CA, USA).

Subsequently, we sequenced and registered the fruiting bodies using isolates cultured from basidiospores. The cellophane tape pieces with attached basidiospores were surface-sterilized with a sodium hypochlorite solution (2% effective chlorine concentration) and then cultured on potato dextrose agar medium with 100 mg/L benomyl for 40 d at 25 °C. The isolated fungi were cultured on potato dextrose agar with 1 mg/L chloramphenicol for 14 days at 23 °C. Genomic DNA was extracted from mycelia collected from the agar surface using the same method as that employed for the fruiting bodies, and DNA yield measurements were performed. A fungal universal primer set was used to amplify the internal transcribed spacer region of rDNA (ITS 1-5.8S rDNA-ITS2) of each strain. The forward and reverse primers were ITS5 (5′-GGAAGTAAAAGTCGTAACAAGG-3′; [25]) and ITS4 (5′-TCCTCCGCTTATTGATATGC-3′; [25]), respectively. PCR was performed using 5 ng of template DNA and 0.25 μM of each of the forward and reverse primers. The PCR conditions were 94 °C for 1 min, followed by 30 cycles of 98 °C for 10 s, 55 °C for 15 s, and 68 °C for 30 s, with a final extension at 68 °C for 7 min. Sequencing was performed using the SeqStudio3 system (Applied Biosystems, Carlsbad, CA, USA).

We were granted access to cut down two decaying *C. pisifera* trees adjacent to plot A. Consequently, heartwood samples were collected from the two cut trees at 1.2 m above ground level using chisels. Conversely, in plots B and C, heartwood samples were collected from internally decayed *C. obtusa* and *L. kaempferi* trees using an Increment borer (Haglöf Sweden AB, Långsele, Sweden). Heartwood samples were collected from 2 trees in plot A, 6 trees in plot B, and 10 trees in plot C with internal decay detected. *S. himantioides* was isolated from all samples in plot A, and from one each of *C. obtusa* and *L. kaempferi* in plots B and C (Appendix A). To assess the occurrence of *S. himantioides* fruiting bodies on the interior of trees, two living trees with fruiting bodies were cut and dismantled. One *Cryptomeria japonica* (Thunb. ex L.f.) D.Don and one *C. obtusa* trees were selected and cut from the area outside plots A–C; these two trees were located in the same forest stand. After dismantling the cut trees, the color of the heartwood was visually recorded to determine whether the fruiting bodies penetrated the bark and reached the sapwood. The heartwood samples were collected from these trees at 0 m and 1.2 m above ground level using a chisel. The samples were sterilized and cultured using the same method as that employed for basidiospores. Subsequently, the DNA was extracted and sequenced; the raw sequence data are available in the International Nucleotide Sequence Database (INSD) (Appendix A).

### 2.3. Statistical Analysis

#### 2.3.1. Occurrence of *S. himantioides* Fruiting Bodies in the Plots

All statistical analyses were performed using the statistical analysis software R (version 4.0.2). To determine the occurrence of *S. himantioides* fruiting bodies in the plots and the occurrence and progression of heartwood decay, the percentages of fruiting bodies and internally decayed wood detected in each survey year and plot were compared using the Bonferroni method [26]. We also compared the percentage of living trees with single-year and multi-year fruiting body occurrence in 2023 using the Fisher test. In addition, visually, the *S. himantioides* fruiting bodies appeared to occur more frequently in the lower slope direction. The Rayleigh test was conducted to examine whether the direction of *S. himantioides* fruiting body development was biased toward the lower slope direction using the *circular* package [27].

#### 2.3.2. Spatial Distribution Patterns of Living Trees with *S. himantioides* Fruiting Bodies

The spatial distribution pattern of living trees with *S. himantioides* fruiting bodies was analyzed using the pair correlation function *g*(*r*), which can be plotted as a graph. The function characterizes the variability of the pattern of tree locations [28], defining the tree density or intensity λ as the mean number of trees per area. Considering an infinitesimally small circle of area d*F*, the probability of finding one tree in it is *λ*d*F*. With two very small circles d*F*_1_ and d*F*_2_, with a distance r between them, the probability of finding a tree in each of the small areas, *P*(*r*), can be expressed as:*P*(*r*) *=* λ^2^·*g*(*r*)·d*F*_1_·d*F*_2_(2)

The function *g*(*r*) is a function of the interpoint distance r. In a forest with trees distributed at random, *g*(*r*) = 1, indicating that the tree locations are spatially uncorrelated. When trees tend to have a regular distribution (e.g., plantation), *g*(*r*) = 0. Values of *g*(*r*) > 1 indicate that the interpoint distances around r are relatively more frequent than those in a forest with random tree locations [28,29,30,31]. If the spatial distribution pattern of all living trees in a plot is not regular, the arrangement of living trees with fruiting bodies will also be irregular. Therefore, we first examined the spatial distribution pattern of all living trees by year and plot, including those with and without fruiting bodies. When the spatial distribution pattern of all living trees was regular, we examined the spatial distribution pattern of living trees with fruiting bodies.

All spatial analyses were performed using the *spatstat* package [32], and Ripley’s isotropic correction [33] was used for edge correction. We used the Monte Carlo method to test the null hypothesis; for bivariate spatial interactions between two groups; the null hypothesis was spatial independence. The estimated function was compared with the theoretical function under the null hypothesis through a test statistic whose expected null-hypothesis value was zero at all distances. We conducted 999 Monte Carlo simulations to generate an envelope curve with a 95% confidence interval to test the significance of the point pattern analysis.

#### 2.3.3. Investigation of the Development Factors of *S. himantioides* Fruiting Bodies

A generalized linear model (GLM) and generalized linear mixed model (GLMM) were used to examine the factors affecting fruiting body development in each tree species and study year. For *C. pisifera*, a GLM with a log-link function and binomial error distribution was fitted, using the presence or absence of *S. himantioides* fruiting bodies as a response variable and *DFr* as an indicator of heartwood decay progression. The DBH and fruiting body heights were used as explanatory variables, which were standardized to determine the strength of their influence. For *C. obtusa* and *L. kaempferi*, a GLMM with a log-link function and binomial error distribution was fitted. The response and explanatory variables were the same as those used for *C. pisifera*, but a plot ID was added as a random effect. Living trees not diagnosed with decay were excluded from all these analyses.

Model selection was performed using the Akaike Information Criterion (AIC) for all possible model combinations. Models with a difference from the best model (ΔAIC) < 2 were adopted [34]. We checked for collinearity among the predictors of the adopted models, applying a cutoff threshold of variance inflation factor (VIF) > 3 according to Zuur et al. [35] to exclude models with these pairs of predictors and avoid multicollinearity. Model variance was also examined with the Pearson’s chi-squared test. The *glmmTMB* package [36] was used to develop the GLM and GLMM, while the *performance* package [37] was used for VIF calculations, and the *DHARMa* package [38] was used for the test of variance.

## 3. Results

### 3.1. Hosts of S. himantioides Fruiting Bodies and Their Locations of Occurrence

In plantations, *S. himantioides* fruiting bodies were found on living trees of *C. japonica*, *C. obtusa*, *C. pisifera,* and *L. kaempferi*, dead trees of the conifers *Abies homolepis* Sieb. & Zucc. and *Clethra barbinervis* Sieb. et Zucc., and on soil on the cut slope of the work road where the roots of *C. pisifera* were exposed. In natural forests, *S. himantioides* fruiting bodies were also found on dead broadleaved trees (Figure 3, Appendix A). The *S. himantioides* fruiting bodies were thin, with a white outer margin and a yellowish-brown fruiting layer that gradually developed from the outer margin. However, the fruiting layers developed on dead *C. barbinervis* and on the soil were dark brown. Most fruiting bodies occurred at the base of trunks or between the main roots, with their lower ends in contact with the ground. These fruiting bodies did not develop underground (Appendix A). In contrast, two *S. himantioides* fruiting bodies with ends not touching the ground were observed (Appendix A).

### 3.2. Color Change of S. himantioides Fruiting Bodies and Spore Production

The fruiting bodies of *S. himantioides* in the plots were yellowish brown from September to October in all survey years, with development of the fruiting layer (Figure 4 and Appendix A). After maturation, the *S. himantioides* fruiting bodies turned dark brown and the fruiting layer lost its ridges, remaining attached to the bark in a thin, dry state. Some fruiting bodies showed temporary white and gray discoloration before and after maturation, respectively. The collected basidiospores were yellowish brown and oval- to kidney-shaped, with an average length of 8.0 ± 0.5 μm (maximum 9.3 μm and minimum 7.0 μm) and average width of 5.5 ± 0.4 μm (maximum 6.5 μm and minimum 4.5 μm).

### 3.3. Isolation and Identification of S. himantioides

PCR results using the *S. himantioides* species-specific primer set were positive for all 117 fruiting bodies collected (Appendix A). Of these, pure cultures of the strains were successfully obtained from basidiospores collected from 92 fruiting bodies. The dismantled living *C. japonica* and *C. obtusa* trees with *S. himantioides* fruiting bodies showed no heartwood discoloration and no fruiting bodies in the interior of the trees. Their fruiting bodies were only present on the bark surface and did not invade the sapwood. In contrast, *S. himantioides* was isolated from heartwood samples collected 0 m above ground level from *C. japonica* trees (Appendix A, Appendix A).

### 3.4. Fruiting Body Development and Heartwood Decay by Plot

The incidence of living trees with *S. himantioides* fruiting bodies in the plots (Table 1) was in the range of 9.1–15.9% in plot A, 26.6–29.5% in plot B, and 36.1–38.9% in plot C. In 2021 and 2022, the incidence of fruiting body development in plots A and C was significantly different (Bonferroni, *p* < 0.05). In 2023, significant differences were observed in the incidence of fruiting body development in plots A and B, as well as in plots A and C (Bonferroni, *p* < 0.01). The number of dead trees in each plot was <10 (Appendix A), with the percentages of dead trees with *S. himantioides* fruiting bodies being 16.7% in plot A (2023 only), 25.6–28.6% in plot B, and 22.2–30.0% in plot C. The incidence of trees with internal decay (Table 2 and Appendix A) was in the range of 52.3–64.4% for plot A, 11.4–13.6% for plot B, and 4.3–17.3% for plot C, with that in plot A being significantly higher than that detected in plots B and C in all study years (Bonferroni, *p* < 0.01). Conversely, no significant differences were observed (Fisher test, *p* > 0.05) in the incidence of living trees with single- and multi-year development of *S. himantioides* fruiting bodies (Appendix A).

The direction of fruiting body development was significantly biased in all plots and survey years, except in 2023 in plot A, with most fruiting bodies developing in the lower slope direction of the trunk (Figure 5).

### 3.5. Spatial Distribution Patterns of Living Trees with S. himantioides Fruiting Bodies

The spatial distribution patterns of living trees in the plots were regularly distributed in a 0–1.5 m range in all plots (Appendix A), suggesting that the distribution of trees with fruiting bodies was not influenced by the initial spatial distribution pattern of living trees. In plot A, the spatial distribution pattern of living trees with *S. himantioides* fruiting bodies was random in all survey years, whereas it was intensive in plots B and C, with two to three separate scales (Figure 6).

### 3.6. Development Factors of S. himantioides Fruiting Bodies by Tree Species

The GLM and GLMM were derived for each tree species excluding living trees that were not included in the decay diagnosis < 60 cm trunk circumference (Appendix A). The DBH and *DFr* of all tree species were included in the selected models, but not the height of the fruiting bodies (Table 3). The VIF between the DBH and *DFr* was < 3, indicating no multicollinearity. The test of variance determined that none of the selected models was overdispersed (*p* > 0.05).

Two models were selected for *C. pisifera* with ΔAIC < 2.00 in 2021 and 2022, while three were selected in 2023. The DBH had a positive effect on the presence of *S. himantioides* fruiting bodies in all survey years, with more fruiting bodies occurring with larger DBH values. Conversely, *DFr* had a negative effect on the presence of *S. himantioides* fruiting bodies in 2022 and 2023, suggesting that more fruiting bodies occur with more advanced decay. However, the absolute value of its standardized partial regression coefficient was smaller than that of DBH (rank 2 model of 2022). Interestingly, *DFr* had a positive effect on the presence of *S. himantioides* fruiting bodies in 2021.

Three models were selected for *C. obtusa* with ΔAIC < 2.00 in all survey years. In 2021 and 2023, the DBH had a positive effect on the presence of fruiting bodies, whereas *DFr* had a negative effect. However, the DBH had a negative effect and *DFr* had a positive effect in 2022, revealing different trends in the presence of fruiting bodies depending on the survey year.

For *L. kaempferi*, three models with ΔAIC < 2.00 were selected in all survey years. The DBH showed a negative effect in 2021 and 2022 but a positive effect in 2023. In contrast, *DFr* showed a negative effect in 2021 and a positive effect in 2022 and 2023, indicating different trends in the presence of *S. himantioides* fruiting bodies depending on the survey year.

## 4. Discussion

### 4.1. Hosts of S. himantioides Fruiting Bodies and Their Locations of Occurrence

Similar to that observed in previous studies [7,8], *S. himantioides* fruiting bodies mainly occurred on coniferous forest trees and soils. While these studies reported the occurrence of *S. himantioides* fruiting bodies on logs and stumps, we were able to unprecedently confirm their occurrence on living trees in this study. As *C. japonica*, *C. obtusa*, and *L. kaempferi*, on which *S. himantioides* fruiting bodies were observed in this study, are major plantation species in Japan, concerns exist regarding the spread of related heartwood decay in Japanese forestry. Moreover, *S. himantioides* fruiting bodies were also observed on dead broadleaf trees in conifer plantations and natural forests. In a previous study, *S. himantioides* was also frequently detected in hardwood (*Quercus rubra* L.) stumps and logs [39], in accordance with the results obtained here.

Most *S. himantioides* fruiting bodies occurred at the base of trunks or between the main roots, with the lower end in contact with the ground. The fruiting bodies of *Serpula lacrymans* (Wulf,: Fr.) Schroet., which are allied to *S. himantioides*, often appear on the lower surface in nature, near the base of trunks or between roots, and can transport water through rhizomorphs connected to the soil [40]. *Heterobasidion* spp. fruiting bodies in Finnish forests have also been reported to predominantly occur in hollows at the base of stumps and dead trees, highlighting the importance of moisture for fruiting body production [16]. Furthermore, Kaarik [41] analyzed the fungal community of decorticated spruce poles standing in soil, observing *S. himantioides* colonies underground with high moisture contents. These findings suggest that moisture conditions play an important role in the development of *S. himantioides* fruiting bodies.

### 4.2. Color Changes of S. himantioides Fruiting Bodies and Spore Production

The color changes of *S. himantioides* fruiting bodies suggest that they did not develop in spring but instead matured to a yellowish-brown color in autumn (September–October), producing basidiospores. This result is in agreement with those of previous studies conducted in Germany and Denmark [7,8]. The shape and size of basidiospores of *S. himantioides* reported in previous research [12] as yellowish brown, oval- to kidney-shaped, and with a size of 8–10.5 µm × 4.6–6 µm, were also in general agreement with the results observed in the present study.

The length of time during which fruiting bodies remain sufficiently intact to distribute spores is a significant feature of fungi ecology [42]. To the best of our knowledge, no previous studies have assessed the longevity of *S. himantioides* fruiting bodies; therefore, the results obtained here may provide important information for characterizing the ecology of this species. Many of the yellowish-brown mature *S. himantioides* fruiting bodies subsequently turned dark brown, with remnants remaining on the bark until the following year. Remnant fruiting bodies of some annual fungi have been found on infected trees that showed fruiting earlier in the season or attached to the trees [43]. These remaining fruiting bodies are important for the detection of *S. himantioides* outside of autumn.

### 4.3. Isolation and Identification of S. himantioides

Some *S. himantioides* fruiting bodies failed to isolate the strains from the basidiospores. The germination and growth of basidiospores in heartwood decay fungi are characterized by very slow germination, with various conditions needing to be met for fungus germination and establishment [44]. Thus, the time of basidiospore production and other environmental conditions may have been limiting factors preventing the isolation of basidiospores from some fruiting bodies.

*Serpula himantioides* was isolated from heartwood samples collected in and around the plots, suggesting that *S. himantioides* may have caused heartwood decay in the three plots. The lower isolation frequency obtained in plots B and C may be attributed to the samples being taken in the Increment borer, which could have a lower probability of collecting heartwood with *S. himantioides* compared to that of sampling by felling.

Previous studies have shown that *S. himantioides* can decay *C. japonica* heartwood in vitro [12], indicating that it could cause heart rot in living *C. japonica*. As no heartwood discoloration was observed in *C. japonica* trees in this study, *S. himantioides* may be present in trees before the onset of heartwood decay or from its early stages. As these fruiting bodies did not invade the heartwood or sapwood of *C. japonica* or *C. obtusa* trees, we suggest that *S. himantioides* fruiting bodies can also occur on healthy trees.

### 4.4. S. himantioides Fruiting Body Development and Heartwood Decay by Plot

Plot A had a low incidence of *S. himantioides* fruiting bodies but advanced heartwood decay, while plots B and C showed a high incidence of fruiting bodies but only early stages of heartwood decay. Schwarze et al. [43] stated that the occurrence of just a single fungus fruiting body may indicate that tree stability is threatened, while noting that fruiting bodies need not necessarily occur every year. In our study, no significant differences were found in the incidence of living trees with single- and multi-year development of *S. himantioides* fruiting bodies, which supports this conclusion.

Several types of dead trees in forest ecosystems contribute to increased habitats for decomposing fungi. This feature may positively impact fungal diversity while increasing the habitat of tree pathogens [39]. In this study, the number of dead trees in the plots was small, so the impact on the habitat of *S. himantioides* and occurrence of heartwood decay in living trees are likely to be small. Furthermore, the bias observed in the developmental orientation of *S. himantioides* fruiting bodies, with the exception of plot A in 2023, suggests the existence of factors that determine the direction of fruiting body development.

### 4.5. Spatial Distribution Patterns of Living Trees with S. himantioides Fruiting Bodies

The spatial distribution pattern of *S. himantioides* fruiting bodies in plots B and C showed an intensive distribution, which may be partially driven by the propagation mode of the fungus. For instance, Gaitnieks et al. [45] noted that the presence of *Heterobasidion* spp. fruiting bodies promotes local infections within a stand via spores deposited near fruiting bodies. *Heterobasidion annosum* (Fr.) Bref. has also been reported to spread from infected to healthy trees via root contact [46], which may promote local infection within a stand. The possibility of infection from the roots has also been suggested for *S. himantioides* [47], but information on the dispersal extent of its basidiospores is lacking. Future studies on the infection of living trees by *S. himantioides* should explore the factors contributing to the intensive distribution of its fruiting bodies.

In contrast, the spatial distribution pattern of living trees with *S. himantioides* fruiting bodies in plot A, where decay progressed, was random. The incidence of living trees with *S. himantioides* fruiting bodies was lower in plot A than that in plots B and C. This result may be one of the reasons why no clear spatial distribution patterns, such as concentrated or uniform distribution, were observed in plot A. This suggests that the spatial distribution pattern of living trees with *S. himantioides* fruiting bodies may change from a concentrated distribution to a random distribution as decay progresses. The distribution of living trees with fruiting bodies is considered uniform once *S. himantioides* has widely spread in the soil. Therefore, the distribution of living trees with fruiting bodies in plot A may have been random because *S. himantioides* was in the process of spreading in the soil.

The spatial distribution pattern of living trees with *S. himantioides* fruiting bodies in each plot did not change significantly throughout the study period. Therefore, the extent to which *S. himantioides* fruiting bodies occur on living trees does not change over a time scale of approximately 2 years. This information could be useful for identifying *S. himantioides* fruiting bodies in forests. Understanding the spatial distribution pattern of fruiting bodies is also useful for estimating the area of distribution of genet and individuals, as well as the timing of their invasion and establishment in forests. For example, Smith et al. [48] identified individuals of *Armillaria gallica* Marxm. & Romagn that occupied at least 15 ha and had been genetically stable for over 1500 years based on fruiting body location and DNA analysis of strains isolated from them. Similarly, the area of distribution of *S. himantioides* individuals and the timing of their entry and establishment in forests may possibly be estimated by analyzing the distribution of the fruiting bodies and isolated strains revealed in this study.

### 4.6. Development Factors of S. himantioides Fruiting Bodies by Tree Species

In *C. pisifera*, DBH had a positive effect on the presence of *S. himantioides* fruiting bodies in all survey years, suggesting that the larger the DBH, the more likely it is that fruiting bodies will occur. The absolute value of the standardized partial regression coefficient of DBH was larger than that of *DFr* in the rank 2 model for 2022, suggesting that DBH had a greater impact than *DFr*. Müller et al. [16] reported that large wood pieces are advantageous for fruiting body development of *Heterobasidion* spp. because they can support the large mycelium required for fruiting body formation and have stable moisture and temperature conditions. This suggests that in *C. pisifera*, larger DBH values offer more favorable conditions for the development of *S. himantioides* fruiting bodies. In *C. obtusa* and *L. kaempferi*, where DBH did not show a consistent effect on the presence or absence of *S. himantioides* fruiting bodies, other factors may have a significant effect on fruiting body formation (e.g., mycelium nourishment conditions, such as deciduous leaves and fallen branches).

No consistent trend was found between *DFr* and fruiting body presence in any of the studied tree species. Müller et al. [16] reported that the ratio between the decayed wood areas and wood disc areas is the most significant variable in the occurrence of fruiting bodies in cull pieces of *Heterobasidion* spp. using a logistic regression analysis. Thus, the results of this previous study differ from those observed here. Although information on heartwood decay and fruiting body development in living trees is insufficient, the development of fruiting bodies regardless of the progress of decay may be an ecological characteristic of *S. himantioides*. This idea is supported by the lack of heartwood discoloration or decay observations in the dismantled *C. japonica* and *C. obtusa* trees with *S. himantioides* fruiting bodies in this study. At present, it is unclear how the fruiting bodies of *S. himantioides*, which develop on undecayed or early decayed trees, obtain their nutrition. Similarly, we cannot ascertain whether they are the same individuals as the *S. himantioides* present in heartwood decay. Recent microsatellite analyses have been performed to investigate the presence or absence of sexual reproduction in *Cryphonectria parasitica* (Murrill) M.E. Barr, which causes chestnut blight [49], and to detect some multilocus genotypes (MLGs) in *Hymenoscyphus fraxineus* (T.Kowalski) Baral, Queloz & Hosoya in one *Fraxinus excelsior* L. tree [50]. Polymorphism analysis using microsatellite markers may be useful for clarifying these issues of *S. himantioides*. This study analyzed the decay of trees with *S. himantioides* fruiting bodies based on the results of the lateral impact vibration method but could not examine the health of trees with *S. himantioides* fruiting bodies. Birkemoe et al. [51] measured carbon, nitrogen, and secondary metabolites in the wood of *Populus tremula* L. and found that nitrogen concentrations significantly affected fungal community composition, and reported secondary metabolites representing an important niche dimension. In the future, by measuring and analyzing secondary metabolites and chemicals, it may be possible to assess the health status of trees with *S. himantioides* fruiting bodies and to clarify the relationship between decay and trees with *S. himantioides* fruiting bodies.

## 5. Conclusions

Based on a two-year-and-six-month study on living trees, we investigated the occurrence of *S. himantioides* fruiting bodies in forest stands with different hosts, development sites, decay progressions, spatial distribution patterns, and factors driving occurrence by tree species. *S. himantioides* fruiting bodies were found on living trees of *C. japonica*, *C. obtusa*, *C. pisifera*, and *L. kaempferi*, which are the main plantation trees in Japan, and predominantly occurred in the lower direction of the slopes. Our results also suggest that *S. himantioides* fruiting bodies may be intensively distributed in forests where heartwood decay has not progressed. Furthermore, no clear relationship was found between the development of *S. himantioides* fruiting bodies and heartwood decay, suggesting that these may also occur in healthy trees and living stands during the early stages of heartwood decay. The results obtained in this study will contribute to a better understanding of the ecological position and survival strategies of *S. himantioides*, which have been largely unexplored to date.

## Figures and Tables

**Figure 1 jof-10-00572-f001:**
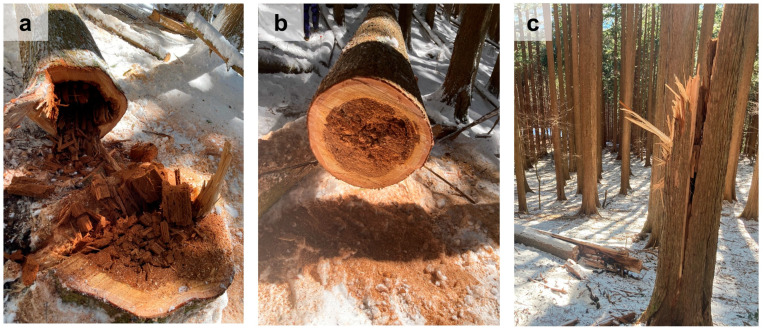
Heart rot caused by *Serpula himantioides* in living trees of *Chamaecyparis pisifera* in Japan. (**a**) Stump of a heart rot tree; (**b**) Transverse section of decayed wood. Brown rot was observed in most of the heartwood. (**c**) A fallen tree caused by heart rot.

**Figure 2 jof-10-00572-f002:**
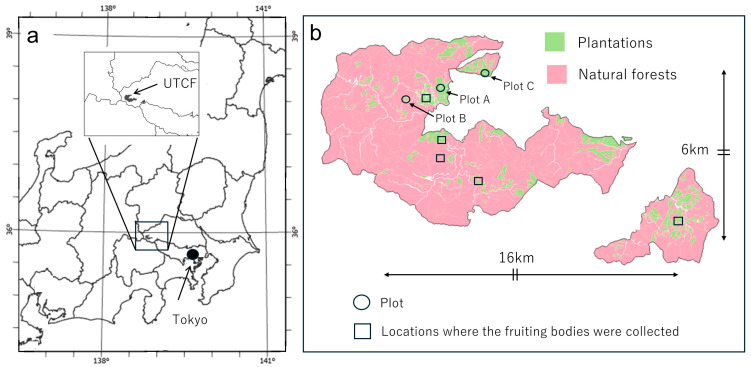
Location of the study site. (**a**) Location of the University of Tokyo Chichibu Forest (UTCF) in Japan. The background map source was obtained from the Geospatial Information Authority of Japan. (**b**) Location of the main *Serpula himantioides* fruiting body collection sites and plots within the UTCF. Arrows indicate the area traversed.

**Figure 3 jof-10-00572-f003:**
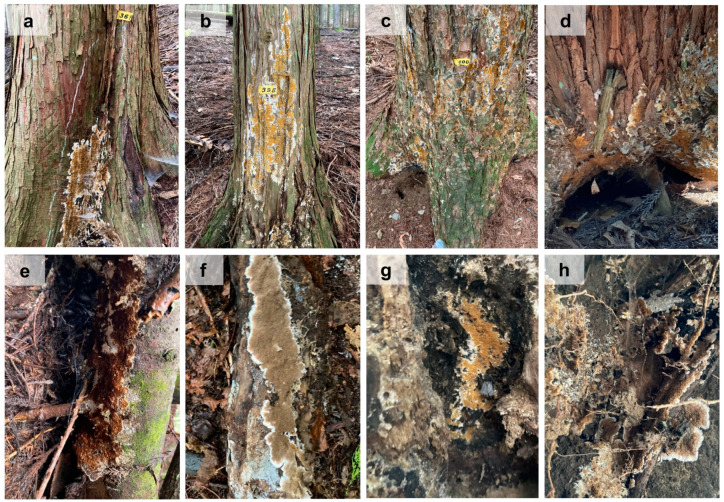
Fruiting bodies of *Serpula himantioides* found in the University of Tokyo Chichibu Forest. Living (**a**) *Chamaecyparis pisifera*; (**b**) *Chamaecyparis obtusa*; (**c**) *Larix kaempferi*; and (**d**) *Cryptomeria japonica* trees. Dead (**e**) *Abies homolepis*; (**f**) *Clethra barbinervis*; and (**g**) broadleaved trees. (**h**) Soil with exposed roots of *Chamaecyparis pisifera*.

**Figure 4 jof-10-00572-f004:**
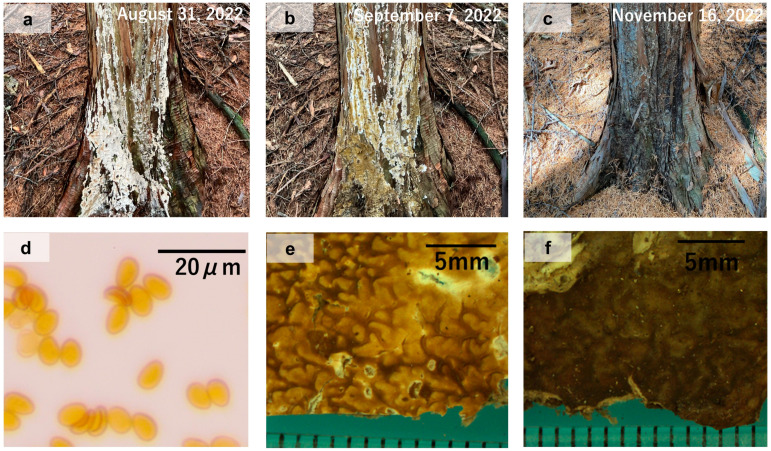
Examples of color variation of *Serpula himantioides* fruiting bodies on living *Chamaecyparis obtusa* trees. (**a**) White fruiting body (31 August 2022); (**b**) Yellowish-brown fruiting body (7 September 2022); and (**c**) Dark-brown fruiting body (16 November 2022), with a discolored upper left part. (**d**) Basidiospores of *Serpula himantioides*; (**e**) Surface of yellowish-brown fruiting body; (**f**) Surface of dark-brown fruiting body.

**Figure 5 jof-10-00572-f005:**
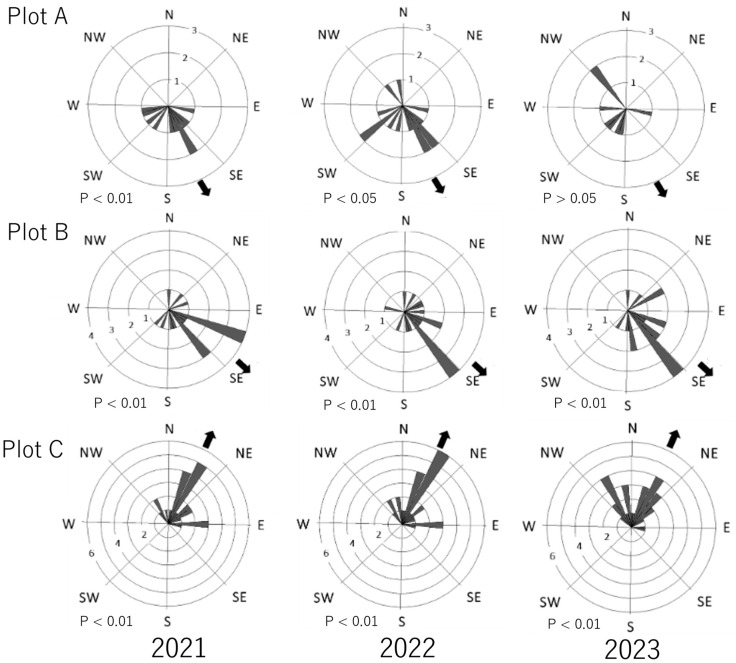
Rose diagram of development orientation of *Serpula himantioides* fruiting bodies in living trees. *p*-values indicate results from the Riley test. Fruiting bodies that were too wide were excluded from the analysis. Arrows indicate the downslope direction. The number and area indicated by the bars are not proportional.

**Figure 6 jof-10-00572-f006:**
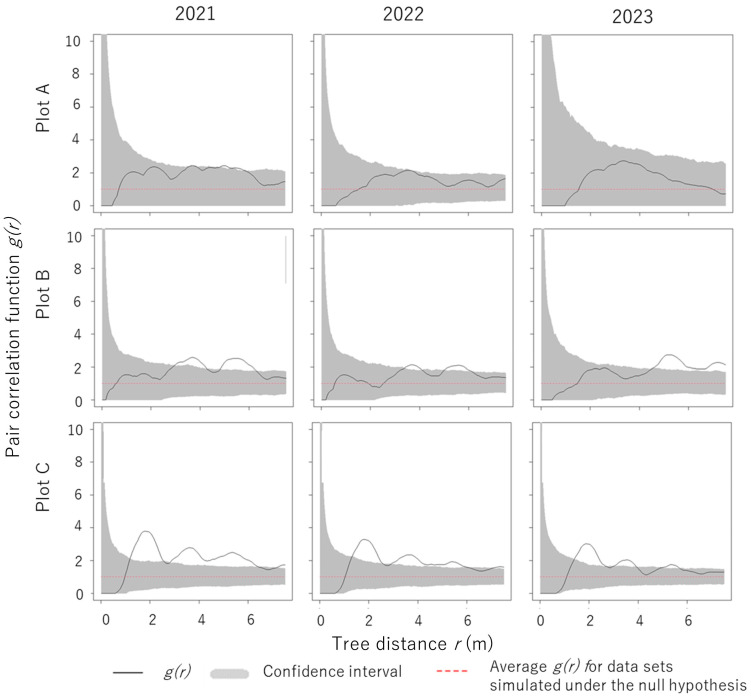
Pair correlation function showing the spatial distribution pattern of living trees with *Serpula himantioides* fruiting bodies. Solid lines show the actual spatial distribution patterns in each plot, gray lines show the 95% confidence interval after 999 Monte Carlo simulations, and dashed lines show the mean values from the simulations. *g*(*r*) values greater than the confidence interval indicate an aggregated population distribution; *g*(*r*) values lower than the confidence interval indicate a uniformly distributed population; and *g*(*r*) values within the confidence interval indicate a randomly distributed population.

**Table 1 jof-10-00572-t001:** Number and incidence of living trees with *Serpula himantioides* fruiting bodies.

Survey Year	2021			2022			2023		
Plot	A	B	C	A	B	C	A	B	C
Number of Living trees									
*Chamaecyparis pisifera*	89	0	0	88	0	0	88	0	0
*C. obtusa*	0	44	51	0	43	51	0	43	51
*Larix kaempferi*	0	29	22	0	18	21	0	17	21
Total	89	64	73	88	61	72	88	60	72
Number of living trees with *Serpula himantioides* fruiting bodies									
*C. pisifera*	11	0	0	14	0	0	8	0	0
*C. obtusa*	0	12	15	0	12	17	0	11	18
*L. kaempferi*	0	5	9	0	6	9	0	6	10
Total	11	17	24	14	18	26	8	17	28
Incidence of living trees with *S. himantioides* fruiting bodies (%)									
*C. pisifera*	12.4			15.9			9.1		
*C. obtusa*		27.3	29.4		27.9	33.3		25.6	35.3
*L. kaempferi*		25.0	40.9		33.3	42.9		35.3	47.6
Total	12.4	26.6	32.9	15.9	29.5	36.1	9.1	28.3	38.9

**Table 2 jof-10-00572-t002:** Number of living trees with internal decay detected and not detected. Trees with a circumference < 60 cm or with a missing trunk were not subject to measurement.

Survey Year	2021			2022			2023		
Plot	A	B	C	A	B	C	A	B	C
Internal decay not detected (A)									
*Chamaecyparis pisifera*	42	0	0	34	0	0	31	0	0
*C. obtusa*	0	21	24	0	22	22	0	23	24
*Larix kaempferi*	0	18	21	0	17	20	0	15	19
Total	42	39	45	34	39	42	31	38	43
Internal decay detected (B)									
*C. pisifera*	46	0	0	53	0	0	56	0	0
*C. obtusa*	0	4	1	0	4	6	0	4	7
*L. kaempferi*	0	2	1	0	1	1	0	2	2
Total	46	6	2	53	5	7	56	6	9
Not subject to measurement									
*C. pisifera*	1	0	0	1	0	0	1	0	0
*C. obtusa*	0	19	26	0	17	23	0	16	20
*L. kaempferi*	0	0	0	0	0	0	0	0	0
Total	1	19	26	1	17	23	1	16	20
Composition of living trees with internal decay detected (B/(A + B)%)									
*C. pisifera*	52.3			60.9			64.4		
*C. obtusa*		16.0	4.0		15.4	21.4		14.8	22.6
*L. kaempferi*		10.0	4.5		5.9	4.8		13.3	9.5
Total	52.3	13.3	4.3	60.9	11.4	14.3	64.4	13.6	17.3

**Table 3 jof-10-00572-t003:** Model selection results for models of variables that explain decay occurrence (generalized linear model and generalized linear mixed model). Models with a difference from the best model (ΔAIC) < 2 and intercept-only model (Null model) are shown.

Tree Species	Survey Year	Rank ^1^	Coefficient			AIC	ΔAIC	VIF	*p*-Value of Underdispersion Test
			Intercept	DBH	*DFr*				
*Chamaecyparis pisifera*	2021	1	−2.14	0.73		65.42	0.00		>0.05
		2	−2.14	0.78	0.12	67.33	1.90	1.20	>0.05
		Null ^2^	−1.95			68.31			>0.05
	2022	1	−1.96	0.99		70.55	0.00		>0.05
		2	−2.07	0.86	−0.51	70.66	1.11	1.04	>0.05
		Null	−1.65			78.77			>0.05
	2023	1	−2.46	0.64		54.42	0.00		>0.05
		2 (Null)	−2.29			55.42	1.01		>0.05
		3	−2.47	0.61	−0.15	56.29	1.88	1.05	>0.05
*C. obtusa*	2021	1 (Null)	−0.58			69.34	0.00		>0.05
		2	−0.58		−0.11	71.19	1.85		>0.05
		3	−0.58	0.01		71.33	1.99		>0.05
	2022	1 (Null)	−0.37			77.00	0.00		>0.05
		2	−0.38	−0.10		78.87	1.88		>0.05
		3	−0.37		0.01	78.99	1.99		>0.05
	2023	1 (Null)	−0.49			80.99	0.00		>0.05
		2	−0.50		−0.25	82.16	1.77		>0.05
		3	−0.49	0.07		82.92	1.93		>0.05
*Larix kaempferi*	2021	1	−0.77	−0.77	−0.77	53.88	0.00	1.12	>0.05
		2	−0.76	−0.68		55.62	1.73		>0.05
		Null	−0.69			57.47			>0.05
	2022	1 (Null)	−0.47			55.97	0.00		>0.05
		2	−0.48	−0.34		56.98	1.01		>0.05
		3	−0.47		0.15	57.77	1.80		>0.05
	2023	1 (Null)	−0.32			55.73	0.00		>0.05
		2	−0.32	0.11		57.62	1.89		>0.05
		3	−0.32		0.06	57.70	1.97		>0.05

Models that include height of fruiting body as an explanatory variable were excluded because it was not selected. ^1^: Rank of models with a ΔAIC of 2 or less in descending order of ΔAIC. ^2^: Intercept only model.

## Data Availability

The original contributions presented in the study are included in the article and Appendix A, further inquiries can be directed to the corresponding author.

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
