# Peer review of "Occurrence and Characteristics of Serpula himantioides Fruiting Bodies on Living Trees in Japan"

_jof, 2024, doi:10.3390/jof10080572_

Round 1
Reviewer 1 Report
- A brief summary
The authors studied the occurrence of Serpula himantioides basidiocarps in relation to heart rot of various tree species in three different Japanese forest stands (Plot A-D). In addition to dead and living trees, basidiocarps were sampled on the forest floor. Serpula himantioides specimens from different trees were isolated and identified with species specific PCR. The basidiocarps occurrence was analysed spatio-temporal, in respect to plot, tree species, appearance and colour in the course of the year, and the assumed localisation in the affected tree trunk, based on data retrieved from a lateral impact vibration analysis of the studied trees. Numerous statistical methods were used to analyse the collected data and the corresponding parameters were calculated. The main result of the study is that the authors assume on the basis of their data that basidiocarps of H. himantioides may occur in healthy living trees as well as in living trees in the early stages of heartwood decay.
- General concept comments
The manuscript is well written and easy to understand.
I suggest to mention the author when naming scientific species for the first time.
Please check that all species names are italicised, e.g. 535, 537 and 548.
The citation of the authors' publications is to an appropriate extent.
The manuscript is clear, relevant for the field and presented in a well-structured manner but Introduction and Material and Methods should be improved.
The conducted methods are well done and the manuscript’s results should be reproducible based on the details given in the methods section.
Apart from the lateral impact vibration analysis, the raw data is available and correct.
I recommend that figure S1. be included in the manuscript.
It is to discuss whether trees with basidiocarps of of Serpula himantioides are really healthy.
Areas of Strengths
Good statistical evaluation of the data.
Number of studied trees
Good photographic documentation of the basidocarps and affecteds host compartments
- Areas of weakness
Introduction:
As this is exclusively an investigation of Serpula himantioides, I suggest that we not only deal with the symptoms of heart rot, but also report something about the associated or brown rot.
As this is an investigation of Serpula himantioides, I suggest that the authors not only deal with the symptoms of heart rot, but also report something about the associated or caused brown rot. The examined trees may also show wood rot or damage caused by H. himantioides without having heart rot.
Results:
Missing (raw) data of the health status of sampled trees apart from assumed internal decay and discoloration.
Line 160: paragraph 2.2.: In Table S2 there are no data information on Japan Sequenced Read Archiv, but on GenBank accession number: ITS
Please add/ change this in 2.2.
Missing or not explicitly indicated results of the lateral impact vibration analysis. In see Material and Methods:
Line 95: The internal decay status of the living trees in the plot was investigated using the lateral impact vibration method [16–21].
TableS1 Please check an lines
- Please find additional specific comments in the attached, annotated pdfs.

Reviewer 2 Report
This research investigated the relationship between occurrence of S. himantioides fruiting bodies and heartwood decay of trees. The fruiting bodies mature time, growth position in the trees, the correlation of heartwood decay of tree with S. himantioides fruiting bodies etc were recorded and analyzed. And no clear relationship was found between the presence or absence of fruiting bodies and heartwood decay. And The suggestion was given that fruiting bodies can occur in healthy living trees as well as in living trees in the early stages of heartwood decay. However, the results were mainly statistical data and short of experiment confirmation. In fact, there is some found that some trees had a low incidence of S. himantioides fruiting bodies but advanced heartwood decay, while some trees showed a high incidence of fruiting bodies but only early stages of heartwood decay. Maybe there were some secondary metabolites or other chemical substances destroyed the structure of woody cell and will cause heartwood decay of trees. The measurement of key substances is necessary. As a whole, this research is an interesting job with good description, but need to add some confirming data.
In my opinion, this manuscript is good in writing and there is not detail comments.
Round 2
Reviewer 1 Report
I am pleased that the authors have adopted the suggestions and thereby improved the manuscript and made it ready for publication.
I have found a few small errors and made suggestions for improvement.
If these are improved, I recommend the publication of the manuscript and to accept it.
Line 49: Decay fungi are divided into brown rot fungi, which preferentially a􀄴ack and rapidly depolymerize cellulose and hemicelluloses, and white rot fungi, which can progressively degrade both carbohydrates and 51
lignin [13, 14].
Do you mean :
wood decay of fungi belonging to Basidiomycota are (fundamentally) divided into....?
Because there are other types of wood decay, for example soft rot of Ascomycota or special ways of wood decay in special species/ genera e.g. Botryobasidium or Fistulina hepatica.
Please do not abbriviate species names in Figure legends: e.g. Figure 4, 5 and 6
Figure 4. Examples of color variation of S. himantioides fruiting bodies on living C. obtusa trees. (a) 292
White fruiting body (August 31, 2022); (b) Yellowish brown fruiting body (September 7, 2022);
(c) Dark brown fruiting body (November 16, 2022), with a discolored upper left part. (d) Basidio- 294
spores of S. himantioides; (e) Surface of yellowish brown fruiting body; (f) Surface of dark brown 295
fruiting body.
Line 455: the current name for A. bulbosa is: Armillaria gallica Marxm. & Romagn., in Boidin, Gilles & Lanquetin, Bull. trimest. Soc. mycol. Fr. 103(2): 152 (1987)
Line 459: Populuse tremula L. please change to: Populus tremula L.
